# High-Molecular-Weight Plasmids Carrying Carbapenemase Genes *bla*_NDM-1_, *bla*_KPC-2_, and *bla*_OXA-48_ Coexisting in Clinical *Klebsiella pneumoniae* Strains of ST39

**DOI:** 10.3390/microorganisms11020459

**Published:** 2023-02-11

**Authors:** Ekaterina S. Kuzina, Angelina A. Kislichkina, Angelika A. Sizova, Yury P. Skryabin, Tatiana S. Novikova, Olga N. Ershova, Ivan A. Savin, Olga E. Khokhlova, Alexander G. Bogun, Nadezhda K. Fursova

**Affiliations:** 1Department of Training and Improvement of Specialists, State Research Center for Applied Microbiology and Biotechnology, Territory “Kvartal A”, 142279 Obolensk, Russia; 2Department of Culture Collection, State Research Center for Applied Microbiology and Biotechnology, Territory “Kvartal A”, 142279 Obolensk, Russia; 3Department of Molecular Microbiology, State Research Center for Applied Microbiology and Biotechnology, Territory “Kvartal A”, 142279 Obolensk, Russia; 4Department of Clinical Epidemiology, National Medical Research Center of Neurosurgery Named after Academician N.N. Burdenko, 125047 Moscow, Russia

**Keywords:** *Klebsiella pneumoniae*, hybrid plasmid, carbapenemase, OXA-48, NDM-1, KPC-2, virulence genes

## Abstract

Background: *Klebsiella pneumoniae,* a member of the ESKAPE group of bacterial pathogens, has developed multi-antimicrobial resistance (AMR), including resistance to carbapenems, which has increased alarmingly due to the acquisition of carbapenemase genes located on specific plasmids. Methods: Four clinical *K. pneumoniae* isolates were collected from four patients of a neuro-intensive care unit in Moscow, Russia, during the point prevalence survey. The AMR phenotype was estimated using the Vitec-2 instrument, and whole genome sequencing (WGS) was done using Illumina and Nanopore technologies. Results: All strains were resistant to beta-lactams, nitrofurans, fluoroquinolones, sulfonamides, aminoglycosides, and tetracyclines. WGS analysis revealed that all strains were closely related to *K. pneumoniae* ST39, capsular type K-23, with 99.99% chromosome identity. The novelty of the study is the description of the strains carrying simultaneously three large plasmids of the IncHI1B, IncC, and IncFIB groups carrying the carbapenemase genes of three types, *bla*_OXA-48_, *bla*_NDM-1_, and *bla*_KPC-2_, respectively. The first of them, highly identical in all strains, was a hybrid plasmid that combined two regions of the resistance genes (*bla*_OXA-48_ and *bla*_TEM-1_ + *bla*_CTX-M-15_ + *bla*_OXA-1_ + *catB* + *qnrS1* + *int1*) and a region of the virulence genes (*iucABCD*, *iutA*, *terC*, and *rmpA2*::*IS110*). Conclusion: The spread of *K. pneumoniae* strains carrying multiple plasmids conferring resistance even to last-resort antibiotics is of great clinical concern.

## 1. Introduction

Healthcare-associated infections (HAIs) are major public health burdens due to higher treatment costs and more complex patient care [1,2]. They are also associated with high morbidity rates, diagnostic uncertainty, and increased mortality [3]. HAIs can be caused by either exogenous or endogenous origins; the agents can be transferred between patients, healthcare workers, contaminated objects, visitors, or various environmental sources [4]. HAIs could be caused by different microorganisms, including bacteria, fungi, and viruses. Among them, the ESKAPE group of bacterial pathogens (*Enterococcus faecium*, *Staphylococcus aureus*, *Klebsiella pneumoniae*, *Acinetobacter baumannii*, *Pseudomonas aeruginosa*, and *Enterobacter* spp.) cause the most clinically significant and antimicrobial-resistant infections [5,6]. ESKAPE pathogens have developed resistance mechanisms against antibiotics that are the last line of defense, including carbapenems; especially, it concerns *K. pneumoniae,* the causative agent of invasive infections of different localizations [7]. Multidrug-resistant (MDR), extensively drug-resistant (XDR), and pandrug-resistant (PDR) *K. pneumoniae* are particularly a problem among neonates, the elderly, and immunocompromised individuals, as well as for the patients of intensive care units (ICUs) [8,9].

Until the last two decades, carbapenems were the drug of choice for the treatment of severe infections caused by MDR-*K. pneumoniae*. But then, the resistance to carbapenems alarmingly increased due to the acquisition of carbapenemase genes located on specific plasmids and mobile genetic elements. *K. pneumoniae* is the most frequently isolated pathogen from patients in ICUs, which became unprotected against infections caused by carbapenem-resistant bacteria [10].

Carbapenemases, enzymes that are able to hydrolyze carbapenems and many other beta-lactams, were divided into two structural groups: metallo-carbapenemases (class B beta-lactamases, e.g., NDM-, VIM-, and IMP-types) and “serine”-carbapenemases (classes A, C, and D beta-lactamases, e.g., KPC-, DHA-, and OXA-48-types) [11]. Carbapenemase-producing *K. pneumoniae* have been widely identified, and bacteria with the co-existence of two or three carbapenemases have been reported during the last decade. For example, KPC-2 and NDM-1 co-producing *K. pneumoniae* isolates have been identified in India, Pakistan, and China [12]. Recently, even hypervirulent *K. pneumoniae* isolates have been described in China [13]. Emergence of *K. pneumoniae* co-producing NDM-1 and OXA-48 carbapenemases have been reported from Turkey, Greece, Italy, Czech, Russia and other countries [14,15,16]. One strain was reported as a carrier of both *bla*_KPC-2_ and *bla*_NDM-1_ genes [17], while the other strain was reported as a carrier of both *bla*_KPC-2_ and *bla*_OXA-48_ genes [18]. *K. pneumoniae* with triple carbapenemase (KPC/NDM-1/OXA-48) genes resistance have been detected in Saudi Arabia [19]. Most recently, a *K. pneumoniae* isolate co-producing three carbapenemases has been collected from a Canadian patient hospitalized in Romania. Whole genome sequencing showed that the *bla*_KPC-2_ gene was located on a 214 Kb IncFIB(K)/IncFII(K) plasmid, the *bla*_NDM-1_ gene-on a 104 Kb IncFIB(pQil)/IncFII(K) plasmid, and the *bla*_OXA-48_ gene-on a 64 Kb IncL plasmid. The uncommon co-carriage of genes encoding different classes of carbapenemases endows them with high carbapenem resistance [20].

Most carbapenemase genes are carried by plasmids, which are extrachromosomal DNA elements that may self-replicate and move horizontally, substantially facilitating their transmission in bacteria [21]. The transmissibility of genes located on mobile elements led to the formation of hybrid plasmids that carry both antimicrobial resistance and virulence (Vir) genes. The plasmid-mediated genetic factors conferring the hypervirulent phenotype include the *rmpA* and *rmpA2* genes, regulators that increase capsule production, and several siderophore gene clusters [22]. The hybrid plasmids have a large size (300–400 Kb) and were formed as a result of recombination. Despite the fact that the virulence potential of isolates harboring hybrid plasmids is not well studied, it could be a new challenge for health care. The recombination and rearrangement of MDR plasmids and virulence plasmids have occurred during evolution. In recent years, many studies have reported the evolution of *K. pneumoniae* into carbapenem-resistant hypervirulent *K. pneumoniae* (CR-hvKP) via the acquisition of hybrid plasmids. Many variants of Vir-KPC plasmids have been described to date in different regions [23]. Emergence of hybrid resistance and virulence plasmids harboring ND metallo-beta-lactamase in *K. pneumoniae* of different STs (ST15, ST147, ST395, and ST874) has been reported in Russia [24]. The *bla*_OXA-48_ carbapenemase gene was identified in a hybrid plasmid harboring an exogenetic chromosome-located fragment and acquired the additional resistance gene *bla*_SHV-12_ due to the recombination of the IS26-like elements in Egypt [25].

In a previous report, we described the emergence of *K. pneumoniae* isolates carrying three carbapenemase genes (*bla*_NDM-1_, *bla*_KPC-2_, and *bla*_OXA-48)_, the cephalosporinase gene (*bla*_CTX-M-15_), and two class 1 integron genes simultaneously in the Moscow neuro-ICU at seven-point prevalence surveys, which included 60 patients in the neuro-ICU. A total of 293 clinical samples were analyzed, including 146 rectal and 147 tracheal swabs, and 344 gram-negative bacteria isolates were collected [26]. In this article, we characterized the complete genomes of four clonally related XDR *K. pneumoniae* carbapenem-resistant clinical strains harboring large hybrid plasmids and determined the location of the resistance and virulence genes.

## 2. Materials and Methods

### 2.1. Bioethical Requirements

The study was a point-prevalence survey. The Burdenko National Medical Research Center of Neurosurgery Review Board approved the study and granted it consent waiver status (approval code: #11/2018, approval date: 1 November 2018). Clinical isolates were labeled without the patients’ names, dates of birth, addresses, disease histories, personal documents, or other personal materials.

### 2.2. Bacterial Strains, Growing, and Identification

Four clinical *K. pneumoniae* strains were isolated from rectal and tracheal swabs collected from four patients of the neuro-ICU of the Burdenko National Medical Research Center of Neurosurgery, Moscow, Russia, in October of 2019. Bacteria were growing in Luria-Bertani (LB) broth (Difco, Rockville, MD, USA) and lactose TTC agar with Tergitol-7 (SRCAMB, Obolensk, Russia) containing 50 mg/L ampicillin (Thermo Fisher Scientific, Waltham, MA, USA) at 37 °C. Bacterial identification was done using the MALDI-TOF Biotyper instrument (Bruker, Karlsruhe, Germany). The following ATCC reference strains were used as controls: *Escherichia coli* ATCC 25922, *Pseudomonas aeruginosa* ATCC 27853, and *Klebsiella pneumoniae* ATCC 700603. Bacterial strains were stored in 15% glycerol at −80 °C.

### 2.3. Susceptibility to Antimicrobials

Antimicrobial susceptibility testing was performed on the Vitek2 Compact system (BioMérieux, Marcy-l’Étoile, Auvergne-Rhône-Alpes, France) using the AST-N101 card (BioMérieux, Marcyl’Étoile, Auvergne-Rhône-Alpes, France). The interpretation was made using the requirements of the European Committee on Antimicrobial Susceptibility Testing (EUCAST), version 12.0, 2022 (http://www.eucast.org) (accessed on 5 September 2022). *Escherichia coli* ATCC 25922 strain was used as quality control. The category of multidrug resistance was determined according to the criteria of Magiorakos et al. [9].

### 2.4. Whole Genome Sequencing, Assembly and Annotation

DNA isolation was performed by the CTAB method [27]. WGS was carried out using the Nextera DNA Library Preparation Kit (Illumina, San Diego, CA, USA) and MiSeq Reagent Kits v3 (Illumina, San Diego, CA, USA) for the Illumina MiSeq platform (Illumina, San Diego, CA, USA). Long reads were obtained using the Rapid Barcoding Kit RBK004 and flowcell R9.4.1 on the MinION platform (Oxford Nanopore, Oxford Science Park, GB). Basecalling was performed with Guppy ver. 5.0.16 (Oxford Nanopore, Oxford Science Park, UK) with default parameters [28]. Short and long raw reads were used to obtain the hybrid assembly of the strain using Unicycler ver. 0.4.7 software (The University of Melbourne, Melbourne, Australia) with default settings that included primary filtering and quality control [29]. Annotation was carried out by the NCBI Prokaryotic Genome Annotation Pipeline (PGAP) ver. 5.3 (National Center for Biotechnology Information, Bethesda, MD, USA) [30].

### 2.5. Whole Genome Analysis

The definition of resistance genes in the complete genome was performed in silico using the online resource ResFinder 2.0 [31] of the Center for Genomic Epidemiology (Technical University of Denmark, Kgs. Lyngby, Denmark). Identification of virulence genes, efflux pump genes, heavy metal resistance genes, and capsule typing was performed in silico using the online resource BIGSdb-Pasteur (https://bigsdb.pasteur.fr/cgi-bin/bigsdb/bigsdb.pl?db=pubmlst_klebsiella_seqdef&page=sequenceQuery, accessed on 5 September 2022). In silico multi-locus sequence typing (MLST) was performed with the online resource MLST 2.0 of the Center for Genomic Epidemiology (Technical University of Denmark, Kgs. Lyngby, Denmark) [32]. The web resource BLAST was used for homologous plasmid searching (National Center for Biotechnology Information, Bethesda, MD, USA) [33]. Whole-genome alignments were performed using Mauve ver. 26 February 2015 [34]. Snippy 4.6.0 software (https://github.com/tseemann/snippy, accessed on 5 September 2022) [35] was used to obtain variant calling into assembled chromosomes and plasmids with default parameters. Easyfig was used as a genome comparison visualizer. The prophage regions in the chromosome were identified by the online resource PHASTER (University of Alberta, Edmonton, AB, Canada) [36].

### 2.6. Nucleotide Sequences Submitted into GenBank Database and Reference Plasmids

Following WGS, four *K. pneumoniae* clinical strains were submitted into the GenBank database:

*K. pneumoniae* strain SCPM-O-B-8912 chromosome (CP086664), plasmid pOXA-48 (CP086665), plasmid pNDM-1 (CP086666), plasmid pIncFIB (CP086667), plasmid pKPC-2 (CP086668), plasmid pColRNAI (CP086669), plasmid p6 (CP086670);

*K. pneumoniae* strain SCPM-O-B-8919 chromosome (CP094991), plasmid pB-8919_OXA-48 (CP094992), plasmid pB-8919_NDM-1 (CP094993), plasmid pB-8919_KPC-2 (CP094994), plasmid pB-8919_ColRNAI (CP094995), plasmid pB-8919_5 (CP094996);

*K. pneumoniae* strain SCPM-O-B-8922 chromosome (CP094363), plasmid pB-8922_OXA-48 (CP094368), plasmid pB-8922_CTX-M-55 (CP094370), plasmid pB-8922_IncFIB (CP094369), plasmid pB-8922_KPC-2 (CP094371), plasmid pB-8922_ColRNAI (CP094372), plasmid pB-8922_6 (CP094373);

*K. pneumoniae* strain SCPM-O-B-8923 chromosome (CP086671), plasmid pB-8923_OXA-48 (CP086672), plasmid pB-8923_NDM-1 (CP086673), plasmid pB-8923_IncFIB (CP086674), plasmid pB-8923_KPC-2 (CP086675), plasmid pB-8923_ColRNAI (CP086676), plasmid pB-8923_6 (CP086677).

For comparative genomic analysis following reference plasmids were used: plasmid unnamed1 of *K. pneumoniae* strain CriePir200 (CP062994), plasmid pNDM-1_Dok01 of *E. coli* strain NDM-1 Dok01 (NC_018994), plasmid pKpQIL of *K. pneumoniae* strain 02288527-42B (MT809701), plasmid pCR14_2 of *K. pneumoniae* strain CR14 (CP015394), plasmid pECO-dc1b of *E. coli* strain ECONIH5 (CP026207), plasmid pE16KP0311-4 of *K. pneumoniae* strain E16KP0311 (CP052623), plasmid p53015-9.3 of *K. pneumoniae* strain 53015_G7 (CP098341).

## 3. Results

### 3.1. Characteristics of K. pneumoniae Strains

Four carbapenem-resistant clinical strains of *K. pneumoniae* SCPM-O-B-8912 (*bla*_OXA-48_, *bla*_NDM-1_, *bla*_KPC-2_, and *bla*_CTX-M-15_), SCPM-O-B-8919 (*bla*_OXA-48_, *bla*_NDM-1_, *bla*_KPC-2_, and *bla*_CTX-M-15_), SCPM-O-B-8922 (*bla*_OXA-48_, *bla*_KPC-2_, and *bla*_CTX-M-55_), and SCPM-O-B-8923 (*bla*_OXA-48_, *bla*_NDM-1_, *bla*_KPC-2_, and *bla*_CTX-M-15_) belonging to the sequence type ST39 and capsular type K-23, were isolated from the rectal (*n* = 3) and tracheal (*n* = 1) swabs of four patients of the neuro-ICU. The patients were admitted to the ICU after neurosurgical operations for various diagnoses. Two patients had additional clinical manifestations of gastrointestinal dysfunction: one patient had both gastrointestinal dysfunction and a respiratory tract infection, and one patient had no such infections but was estimated to be a carrier of the antibiotic resistance genes (Table 1).

According to Magiorakos et al.’s [9] criteria, four strains were attributed to the XDR category; they are resistant to 6–7 functional groups of antimicrobials. All strains were resistant to beta-lactams, nitrofurans, fluoroquinolones, sulfonamides, aminoglycosides, and tetracyclines; one strain was additionally resistant to chloramphenicol; and one strain was resistant to colistin (Table 2).

### 3.2. Characteristics of the Chromosoms

The complete genome assemblies of four *K. pneumoniae* strains contained each one chromosome and the sets of the plasmids. The chromosome sizes in the strains were comparable, and the pairwise distance between them showed a high degree of homology (99.99958–99.99943%). The total number of genes was the same in all the strains (Table 3).

The chromosomes of all four strains carried the *bla*_SHV-11_ non-extended spectrum beta-lactamase (non-ESBL) gene and the *fosA* gene conferring the resistance to fosfomycin, as well as the *oqxA* and *oqxB* efflux pump genes, which are responsible for the resistance to chloramphenicol, nalidixic acid, ciprofloxacin, trimethoprim, benzylkonium chloride, and cetylpyridinium chloride. The point mutations associated with high-level fluoroquinolone resistance were found in the *gyrA* gene (T247A and C248T determining the amino acid substitution Ser83Ile, and G259A—the amino acid substitution Asp87Asn)*,* and in the *parC* gene (G239T defining an amino acid substitution Ser80Ile). Moreover, genes of 5 efflux systems, *acrABR, marAR, soxSR, ramA,* and *oqxABR,* and genes of 5 regulators, *rob, sdiA, fis, envR,* and *rarA,* were detected in the chromosomes of all strains.

Analysis of virulence genetic determinants revealed the *mrkABCDFIJ* gene cluster coding type 3 fimbrial adhesine, the *irp* gene cluster of yersiniabactin biosynthesis, the *ybtAEPQSTUX* gene cluster of yersiniabactin biosynthesis, and the *fyuA* gene encoding the siderophore yersiniabactin receptor—in the chromosomes of all strains.

A comparative analysis of the chromosomes of four *K. pneumoniae* strains revealed ten main structural differences: eight insertions of IS-like elements, one repeat, and one inversion of three genes in the prophage region:-the lysophospholipid acyltransferase family protein gene was functional in the strain SCPM-O-B-8922 (KIF64_00650), but disrupted by insertion of an IS5-like element ISKpn26 family transposase in the strains SCPM-O-B-8912, SCPM-O-B-8919, and SCPM-O-B-8923 (KFX87_000655, KF986_000655, and KIF80_000655, respectively);-the O-antigen ligase family protein gene in the strain SCPM-O-B-8922 (KIF64_00770) was disrupted by insertion of an IS5-like element ISKpn26 family transposase (KIF64_00765), unlike this gene in the other three strains;-the gene *glpT* encoding glycerol-3-phosphate transporter in the strain SCPM-O-B-8923 (KIF80_007475) was disrupted by the insertion of an IS4-like element by an ISVsa5 family transposase;-the insertion of an IS5-like element ISKpn26 family transposase (KIF64_08135) was detected in the intergenic space between the phosphatase PAP2 family protein gene (KIF64_08130) and hypothetical protein gene (KIF64_08140) in the strain SCPM-O-B-8922, in contrast to other three strains;-four genes, homologues to *gmlABC* (providing the structural modification of D-galactan I) and *kfoC* (unknown function), in the *rfb* cluster were deleted and replaced by the insertion of an IS5-like element ISKpn26 family transposase in the strain SCPM-O-B-8922; the *rfb* cluster in the chromosomes of the other three strains was complete and homologous to the *K. pneumoniae* type O1/O2-antigen gene cluster;-the IV secretion system protein gene was disrupted by the insertion of an IS5-like element, ISKpn26 family transposase (KFX87_008725), in the strain SCPM-O-B-8912;-the aspartate/glutamate racemase family protein gene was functional in the strain SCPM-O-B-8922 (KIF64_08860), while it was disrupted by the insertion of an IS1-like element by the IS1B family transposase in the strains SCPM-O-B-8912, SCPM-O-B-8919, and SCPM-O-B-8923 (KFX87_008890, KF986_008880, and KIF80_008890, respectively);-the PhoP/PhoQ regulator membrane protein MgrB gene was disrupted by the insertion of an IS5-like element ISKpn26 family transposase (KIF64_09410) in the strain SCPM-O-B-8922;-the inversion of three genes, KF986_013800, KF986_013805, and KF986_013810, encoding hypothetical proteins, has been detected in the strain SCPM-O-B-8919.

### 3.3. Characteristics of the Plasmids

Six incompatibility groups of plasmids were discovered in the genomes of four *K. pneumoniae* strains. Three strains contained six plasmids, while one strain contained five. Three strains, SCPM-O-B-8912, SCPM-O-B-8919, and SCPM-O-B-8923, contained three large plasmids of Inc-groups IncHI1B, IncC, and IncFIB carrying the carbapenemase genes of three types, *bla*_OXA-48_, *bla*_NDM-1_, and *bla*_KPC-2_, respectively. One strain, SCPM-O-B-8922, contained two named plasmids: IncHI1B, carrying the *bla*_OXA-48_ carbapenemase gene, and IncFIB, carrying the *bla*_KPC-2_ carbapenemase gene; additionally, the IncC-group plasmid carried the *bla*_CTX-M-55_ cephalosporinase gene. Three *K. pneumoniae* strains, SCPM-O-B-8912, SCPM-O-B-8922, and SCPM-O-B-8923, contained the additional large plasmid of the IncFIB/IncFII group, carrying the genes conferring resistance to heavy metals as well as the genes of the toxin-antitoxin system and a conjugative transfer system. Moreover, all four strains carried two small, cryptic plasmids identified as ColRNAI and pUN1 (Figure 1).

#### 3.3.1. Characteristics of IncHI1B Plasmids Carrying the *bla*_OXA-48_ Carbapenemase Gene

All four *K. pneumoniae* strains carried the *bla*_OXA-48_ carbapenemase gene on the large hybrid IncHI1B plasmids (323,074–327,685 bp), which are identical by 99.9% among them. A deletion of nucleotide sequence carrying 10 genes, including the EAC protein gene, the class I ribonucleotide reductase gene, the type II toxin-antitoxin system RelE/ParE genes, and hypothetical protein genes, was detected in the plasmids pB-8919_OXA-48, pB-8922_OXA-48, and pB-8923_OXA-48, compared to the plasmid pB-8912_OXA-48. Moreover, the inversion of the ~112 Kb region was identified in the plasmid pB-8923_OXA-48 compared to the plasmids in the other three strains (Figure 2A).

The hybrid nature of the IncHI1B plasmids was postulated because they were derived from regions that carried both antimicrobial resistance (AMR) genes and virulence genes. Epidemically significant antibiotic resistance genes identified in these plasmids conferring resistance to aminoglycosides, *ant(2″)-Ia*, *aac(6′)-Ib-cr*, and *aadA1*; to quinolones, *qnrS1*; to beta-lactams, *bla*_OXA-1_ and *bla*_TEM-1_; to cephalosporins, *bla*_CTX-M-15_; to carbapenems, *bla*_OXA-48_; to sulfamethoxazole, sul1; and to chloramphenicol, *catA1* and *catB3*. Moreover, *K. pneumoniae* hypervirulence genetic determinants, the *iucABCD* (aerobactin-producing operon), the *iutA* (iron trapping receptor gene), the *terC* (tellurium ion resistance protein gene), and the *rmpA2* gene disrupted by insertion of an IS110 family transposase were identified in these plasmids (Figure 2).

A BLAST search revealed one homologous hybrid plasmid in the GenBank database, recently submitted from the genome of *K. pneumoniae* strain CriePir200, isolated from the sputum of a patient at the N.I. Pirogov National Medical and Surgical Center in Moscow in 2018 and described by Shelenkov et al., 2020 [37]. The genetic environments of the *bla*_OXA-48_ carbapenemase gene were identical in four plasmids in our study, and in the plasmid pCriePir200 (Figure 2A,B). The genetic environments of the *bla*_CTX-M-15_ and *bla*_OXA-1_ beta-lactamase genes were identical in four plasmids in our study, including the presence of additional resistance genes (*bla*_TEM-1_, *catB3*, *aac(6′)-Ib-cr*, and *qnrS1*), but such loci were missed in the plasmid CriePir200 (Figure 2C). As for the virulence gene region, the *rmpA2* gene has not been disrupted by the IS-element in the plasmid pCriePir200, in contrast to four other plasmids in our study (Figure 2B).

#### 3.3.2. Characteristics of IncC plasmids Carrying the *bla*_NDM-1_ Carbapenemase Gene

Three *K. pneumoniae* strains, SCPM-O-B-8912, SCPM-O-B-8919, and SCPM-O-B-8923, contained the IncC plasmid, which carried the *bla*_NDM-1_ carbapenemase gene. Additionaly, the *bla*_OXA-10_ and *bla*_OXA-1_ beta-lactamase genes, as well as genes conferring the resistance to macrolides (*msr(E)* and *mph(E)*), aminoglycosides (*aac(3)-IIa*, *aac(6’)-Ib-cr*, *aadA1*, *aadA2*, and *armA*), sulfamethoxazole (*sul1*), trimethoprim (*dfrA12*), chloramphenicol (*cmlA1* and *catB3*), and rifampicin (*ARR-3*) were located on the IncC plasmids. The disinfectant resistance gene (*qacE*) conferring resistance to quaternary ammonium compounds was determined in these plasmids also. The plasmid sequences were almost identical except for a few single nucleotide polymorphisms (SNPs). The *bla*_NDM-1_ gene is associated with the IS91 insertion sequences. The oxacillinase gene *bla*_OXA-10_ is located downstream of the IS110 insertion sequence as a part of the cassette array of the class 1 intergron; it is, however, associated with the Tn3 transposon. The class 1 integron is flanked by the IS91 insertion sequences and carries the set of gene cassettes conferring resistance to antibacterials. A comparison of the IncC plasmids of our study with the closest homologous plasmid, pNDM-1_Dok01of (query cover 90%, identity 99.9%), identified in *E. coli* ST38 collected in Japan, shows multiple rearrangements in the genetic environment of the *bla*_NDM_, *bla*_OXA-10_, and *bla*_OXA-1_ beta-lactamase genes (Figure 3).

#### 3.3.3. Characteristics of IncFIB(pQil) Plasmids Carrying the *bla*_KPC-2_ Carbapenemase Gene

All *K. pneumoniae* strains in our study contained the IncFIB(pQil) plasmids carrying the *bla*_KPC-2_ carbapenemase gene. The comparison of the plasmids revealed identical structures and the same sequences with a few SNPs. Additional genes for resistance determinants were not found. A comparison of the plasmid pB-8919_KPC-2 with the homologous plasmid pKpQIL of *K. pneumoniae* strain 02288527-42B (query cover 98%, identity 99.6%) revealed the detailed structure of transposon Tn4401 carrying the *bla*_KPC-2_ carbapenemase gene. Unlike the reference plasmid pKpQIL, the mercuric resistance gene cluster was missed in the plasmid pB-8919_KPC-2 (Figure 4).

#### 3.3.4. Characteristics of IncC Plasmids Carrying the *bla*_CTX-M-55_ Cephalosporinase Gene

One *K. pneumoniae* strain in our study, SCPM-O-B-8922, contained the IncC plasmid (169,151 bp) and carried the *bla*_CTX-M-55_ cephalosporinase gene. The closest homologous plasmid in the GenBank database was identified, pECO-dc1b (82% overlap, 99% identity), in the genome of an *E. coli* strain isolated from a hospital water supply in the United States in 2015, but this plasmid doesn’t carry any *bla*_CTX-M_ genes. So, we compared the plasmid pB-8922_CTX-M-55 with the plasmid pCR14_2 (79% overlap, 99% identity) of *K. pneumoniae* strain CR14 carrying the *bla*_CTX-M-2_ gene. The plasmids have a similar structure except for the region of gene resistance cassettes. The plasmid pB-8922_CTX-M-55 contained the genetic determinants of resistance to aminoglycosides (*aac(6’)-Ib-cr*, *aadA1*, and *armA*), sulfamethoxazole (*sul1*), trimethoprim (*dfrA1*), and quaternary ammonium compounds (*qacE*) (Figure 5).

#### 3.3.5. Characteristics of IncFIB(K)/IncFII(K) Plasmids

Three K. pneumoniae strains, SCPM-O-B-8912, SCPM-O-B-8922, and SCPM-O-B-8923, contained the plasmid of the IncFIB(K)/IncFII(K) group (171,935–173,263 bp). These plasmids carried no genes of antibiotic resistance but genes of copper and arsenical resistance, the sil cation-efflux system that confers resistance to silver, the type II toxin-antitoxin system for the RelE/ParE family toxin, Fe(3+) dicitrate transport, and the type-F conjugative transfer system. The sequences of the three plasmids were 90% identical; the difference in size was associated with the IS110 and IS5 copies. A BLAST search for homologous plasmids showed a lot of very similar plasmids in the GenBank database.

#### 3.3.6. Characteristics of Low-Molecular-Weight Plasmid pColRNAI

All K. pneumoniae strains carry the identical pColRNAI plasmids (9294 bp). These plasmids carried the genes encoding colicin, cloacin, and the toxin-antitoxin system RelE/ParE. BLAST search for homologous plasmids to pColRNAI revealed a few similar plasmids of the same sizes, such as the plasmid pE16KP0311-4 (CP052623) of *K. pneumoniae* strain E16KP0311, isolated from the patient’s blood in South Korea, and the plasmid p53015-9.3 of *K. pneumoniae* strain 53015_G7 (CP098341) obtained from *Homo sapiens* in USA.

#### 3.3.7. Characteristics of Low-Molecular-Weight Plasmid pUN1

The small, identical pUN1 plasmids (4352 bp) were identified in the genomes of all *K. pneumoniae* strains in our study. The closest match (95% overlap, 89.94% identity) in the GenBank database was a cryptic plasmid, pSTW0522-30-5 (AP022404), of *Citrobacter portucalensis* isolated from hospital sewage in Japan.

## 4. Discussion

The number of infections caused by antibiotic-resistant bacteria is constantly increasing, and antibiotic resistance, especially to carbapenems, has already become a dangerous problem for health care systems around the world. The carbapenems were the last-line class of antibiotics used for the treatment of infections caused by multi-resistant bacteria, but they lost this position after the emergence and rapid dissemination of carbapenemase-producing agents of HAIs. The spread of the carbapenemase genes was associated with intra- and interspecies plasmid-mediated transfer. The plasmids of several incompatibility groups and different backbones were involved in the rapid dissemination of carbapenem-resistant (CR) *Enterobacteriaceae* [38,39]. The spread of carbapenemases via plasmids has been documented in Russia [24,40].

In this study, we described in detail the plasmids identified in XDR *K. pneumoniae* clinical strains belonging to sequence type ST39 and capsular type K-23, which were isolated at a point-prevalence study from four patients in a neuro-ICU in Moscow; two of them died [26]. Previously, strains of the same genetic line were discovered in Moscow ICUs [16,41]. The novelty of this study is the first report of three plasmids carrying carbapenemase genes of three types, *bla*_OXA-48_, *bla*_NDM-1_, and *bla*_KPC-2_, being expressed simultaneously in bacterial cells on the high-molecular-weight plasmids of the IncHI1B (~326 Kb), IncC (~175 Kb), and IncFIB(pQil) (~102 Kb) incompatibility groups, respectively Mataseje et al. recently described the first complete genome sequence of a *K. pneumoniae* isolate carrying the three carbapenemases listed above in a strain isolated from a Canadian patient hospitalized in Romania [20]. In contrast to the plasmid’s genetic background presented in our study, the *bla*_OXA-48_, *bla*_NDM-1_, and *bla*_KPC-2_ carbapenemase genes were located on the plasmids of IncL (64 Kb), IncFIB(pQil)/IncFII(K) (104 Kb), and IncFIB(K)/IncFII(K) (214 Kb) incompatibility groups, respectively; the strain was attributed to the ST147 cluster that is endemic in Romania. Therefore, we can conclude that there are parallel processes of the formation of multi-resistant, multi-plasmid strains in different regions of the world based on different genetic lines of *K. pneumoniae*.

Importantly, one of the studied plasmids, carrying the *bla*_OXA-48_ carbapenemase gene, contains not only a large set of AMR genes conferring resistance to aminoglycosides, quinolones, beta-lactams, cephalosporins, carbapenems, sulfamethoxazole, and chloramphenicol, but, additionally, the region of virulence genes inherent to hypervirulent *K. pneumoniae: iucABCD, iutA, terC,* and *rmpA2*::*IS110*. This hybrid plasmid has a large size (~326 Kb) and was formed as a result of an unknown event of recombination. In recent years, many studies have reported carbapenem-resistant hypervirulent *K. pneumoniae* (CR-hvKP) [23,24].

Another important aspect of this study is the carriage of multidrug-resistant strains without clinical manifestations of infection. One of the strains described in the study, which carried three carbapenemase genes, was isolated from a patient without HAI. Aside from human-to-human transmission, the hospital environment is a major source of antibiotic-resistant pathogen transmission and a breeding ground for novel resistance combinations. Evolutionary changes and swapping of genetic information can occur both within populations of nosocomial infections on different surfaces and within the organisms of patients. This statement could be confirmed by detection of plasmid-mediated transfer of resistance genes on an IncFIB plasmid between *K. pneumoniae* isolates [42]. In comparison to outbreak investigations, the carrying of CR isolates is studied infrequently. The strategy of infection control required to reduce nosocomial infections, except for careful cleaning measures and monitoring of the microbiomes of hospital environments, should be the identification of ICU patients colonized by CR bacteria [43].

It was shown in this research that the four studied *K. pneumoniae* strains were undoubtedly closely related; their nucleotide sequences of chromosomes were almost identical; only insertions of IS-elements caused the differences. We defined that the strains carried various sets of plasmids, but the nucleotide sequences of the homologous plasmids were very similar. This could point to the spread of clonally related strains, particularly in the ICU, and the acquisition of differences through continuous evolution. The molecular and genomic study of bacterial pathogens is currently focused on tracking clonally evolving lineages. While surveillance of epidemic clones is important, plasmids can spread horizontally between strains and even species. The plasmids are the primary carriers of antibiotic resistance genes across many pathogens, and they should be analyzed more accurately with high-resolution techniques [44].

In research studies, it has been shown that the spread of the carbapenemase gene occurs across a small number of high-risk clones [45]. In our study, *K. pneumoniae* strains belonging to ST39 and K-23, which were rarely isolated from patients and poorly presented in the GenBank database, were able to accumulate plasmids carrying three carbapenemase genes, *bla*_NDM-1_, *bla*_KPC-2_, and *bla*_OXA-48_, simultaneously. A detailed analysis of the genetic environments of these genes revealed the common structure of these regions. Certainly, the progenitor of these strains had the ability to accept the plasmids. Based on the fact that exact copies of the plasmids carrying *bla*_OXA-48_, *bla*_NDM-1_, and *bla*_KPC-2_ genes described in our study were not found in the GenBank database, it can be assumed that the plasmids have high rates of recombination and rearrangements; the structural variability is changing over time. Because plasmids can undergo rapid evolutionary change, plasmid transmission events may go undetected if only general comparisons of the strain’s genomes are performed. We anticipate the need for additional comparisons of the nucleotide diversity of the plasmids and comparative analysis. A detailed study of hospital strain genomes could identify the source of HAI outbreaks. The large-scale surveillance could reveal clinical and biological insights pertaining to the hospital microbiome as a reservoir of pathogens.

## 5. Conclusions

In conclusion, the spread of *K. pneumoniae* strains carrying multiple plasmids conferring resistance even to last-resort antibiotics is of great clinical concern. Here we described three high-molecular-weight plasmids carrying the *bla*_OXA-48_, *bla*_NDM-1_, and *bla*_KPC-2_ carbapenemase genes in the strains of ST39, capsular type K-23, that were connected with high mortality. One of the plasmids, p_OXA-48, was a hybrid plasmid, carrying the genetic determinants of hyper-virulence, in addition to the large set of AMR genes. At the same time, one of the strains was associated with intestinal carriage in a patient without clinical manifestations of infection. The above indicates that continuous monitoring of high-risk clones and the detection of resistance mechanisms are necessary strategies to combat such threats. Further investigations are needed to fully understand the epidemiology of *K. pneumoniae* strains co-producing multiple molecular mechanisms of resistance, especially based on combining the results of whole genome sequencing, transcriptome analysis, and clinical characteristics.

## Figures and Tables

**Figure 1 microorganisms-11-00459-f001:**
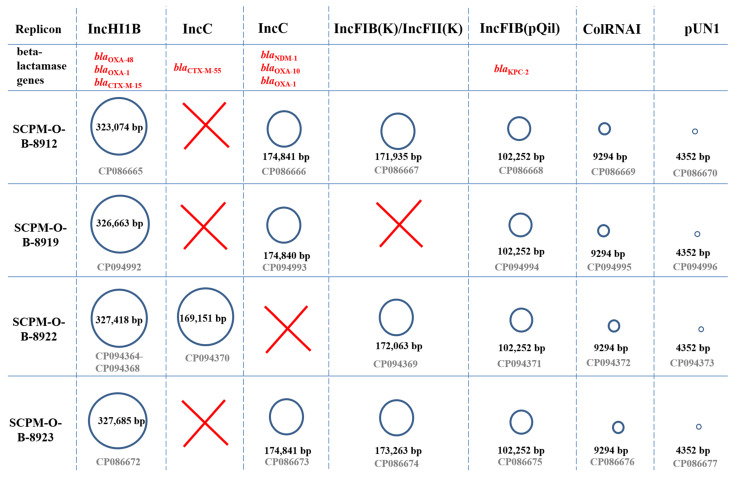
Plasmids identified in K. pneumoniae strains SCPM-O-*B*-8912, SCPM-O-B-8919, SCPM-O-B-8922, and SCPM-O-B-8923.

**Figure 2 microorganisms-11-00459-f002:**
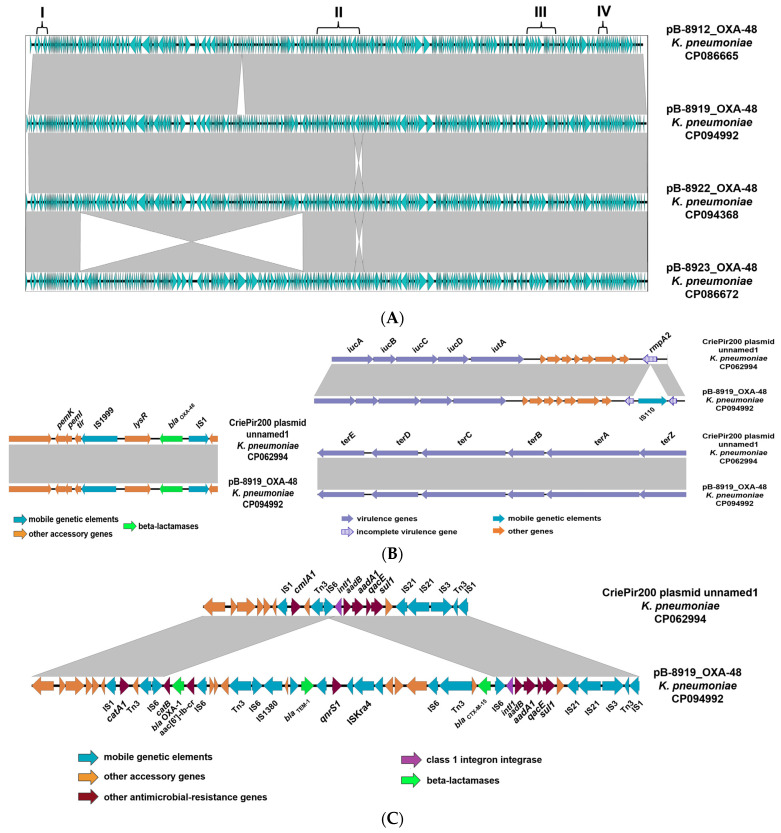
Easyfig comparison: (**A**) IncHI1B plasmid in four *K. pneumoniae* clinical strains carrying carbapenemase gene *bla*_OXA-48_ (region I), integron class 1 and beta-lactamase genes *bla*_TEM-1_, *bla*_CTX-M-15_, and *bla*_OXA-1_ (region II), virulence genes *iucABCD, iutA, and rmpA2* (region III), and virulence genes *terEDCDAZ* (region IV); (**B**) genetic environments of the *bla*_OXA-48_ gene (region I) and virulence genes (regions III and IV); (**C**) genetic environments of beta-lactamase genes *bla*_TEM-1_, *bla*_CTX-M-15_, and *bla*_OXA-1_ (region II) in the plasmid pB-8919_OXA-48 of *K. pneumoniae* strain SCPM-O-B-8919 compared to the plasmid unnamed 1 of *K. pneumoniae* strain CriePir200 [37] The arrows show the open reading frames (*orfs*) and their orientation.

**Figure 3 microorganisms-11-00459-f003:**
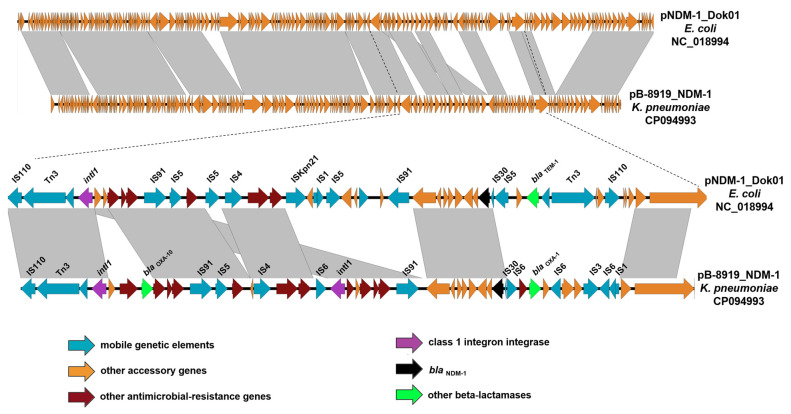
Easyfig comparison of IncC plasmids pB-8919_NDM-1 (*K. pneumoniae* strain SCPM-O-B-8919), pNDM-1_Dok01 (*Escherichia coli* strain NDM-1 Dok01), and the genetic environment of the genes *bla*_NDM-1_, *bla*_OXA-10_, and *bla*_OXA-1_. The arrows show the *orfs*, their length, and their orientation.

**Figure 4 microorganisms-11-00459-f004:**
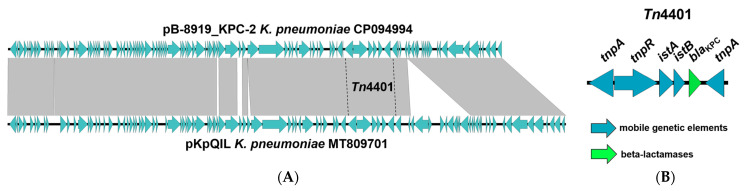
Easyfig comparison of IncFIB(pQil) plasmids (*K. pneumoniae* strain SCPM-O-B-8919 and *K. pneumoniae* strain 02288527-42B) (**A**). structure of transposon (Tn4401) with location of carbapenemase gene *bla*_KPC-2_ (**B**). The arrows show the *orfs*, their length, and their orientation.

**Figure 5 microorganisms-11-00459-f005:**
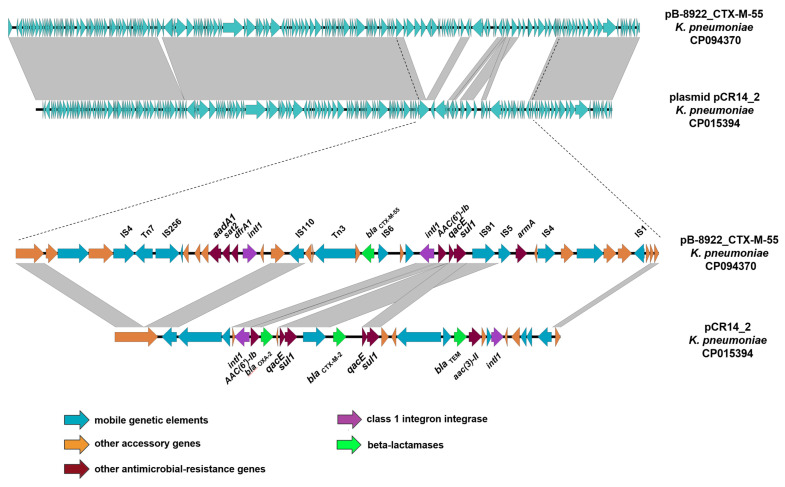
Easyfig of pB-8922_CTX-M-55 and plasmid pCR14_2 with the detailed genetic environment of the gene *bla*_CTX-M-55_. The arrows show the *orfs*, their length, and their orientation.

**Table 1 microorganisms-11-00459-t001:** Patient’s information.

Information	SCPM-O-B-8912	SCPM-O-B-8919	SCPM-O-B-8922	SCPM-O-B-8923
Patient (gender, age)	Male, 36	Male, 67	Male, 59	Female, 64
ICU state, days	42	23	95	96
Outcome	Discharged	Discharged	Fatal	Fatal
Isolation source	rectal swab	rectal swab	rectal swab	tracheal swab
Collection date	18 October 2019	21 October 2019	21 October 2019	21 November 2019
Patient’s diagnosis	Diffuse cerebral and cerebellar injury, unspecified	Secondary malignant neoplasm of the brain and cerebral meninges	Nontraumatic intracerebral hemorrhage in the hemisphere, unspecified	Benign neoplasm of cranial nerves
Infectious manifestation	GD	No	GD	GD, RTI

Note: GD, gastrointestinal dysfunction; RTI, respiratory tract infection; No, there will be no GD or RTI.

**Table 2 microorganisms-11-00459-t002:** Antimicrobial susceptibility of *K. pneumoniae* strains (MIC, mg/L).

Antimicrobials	SCPM-O-B-8912	SCPM-O-B-8919	SCPM-O-B-8922	SCPM-O-B-8923
Ampicillin	>16	>16	>16	>16
Amoxicillin-clavulanic acid	>128	>128	>128	>128
Cefoperazone	64	64	32	64
Cefotaxime	>32	>32	>32	>32
Ceftazidime	>256	>256	>256	>256
Cefepime	>16	>16	>16	>16
Aztreonam	32	64	64	32
Imipenem	>64	>64	>64	>64
Meropenem	128	>256	>256	256
Ertapenem	>4	>4	>4	>4
Tigecycline	>4	>4	4	>4
Ciprofloxacin	256	256	256	>256
Chloramphenicol	8	8	4	16
Gentamicin	>256	>256	>256	>256
Netilmicin	>16	>16	>16	>16
Amikacin	>32	>32	>32	>32
Trimethoprim-sulfamethoxazole	>160	>160	>160	>160
Fosfomycin	128	128	>128	128
Colistin	≤0.5	≤0.5	>8	≤0.5

**Table 3 microorganisms-11-00459-t003:** Major genome characteristics of clinical *K. pneumoniae* strains.

Features	SCPM-O-B-8912	SCPM-O-B-8919	SCPM-O-B-8922	SCPM-O-B-8923
GenBank chromosome	CP086664	CP094991	CP094363	CP086671
Chromosome size, bp	5,351,360	5,350,432	5,348,898	5,351,820
Genes (total)	6079	5902	6079	6091
Genes (coding)	5788	5626	5790	5801
Genes (RNA)	126	126	126	126
rRNA genes (5S, 16S, 23S)	9, 8, 8	9, 8, 8	9, 8, 8	9, 8, 8
tRNA genes	88	88	88	88
Pseudo Genes (total)	165	150	163	164
- frameshifted	67	61	65	66
- incomplete	102	88	101	102
- internal stop	25	22	25	26
- multiple problems	24	18	23	25
Plasmids	6	5	6	6

## Data Availability

The plasmids sequences for *K. pneumoniae* have been deposited in GenBank under accessions no. (CP086665, CP086666, CP086667, CP086668, CP086669, CP086670, CP094992, CP094993, CP094994, CP094995, CP094996, CP094364-CP094368, CP094370, CP094371, CP094372, CP094373, CP086672, CP086673, CP086674, CP086675, CP086676, and CP086677), the chromosomes sequences have been deposited in GenBank under accessions no. (CP086664, CP094991, CP094363, and CP086671). The raw sequence MinION reads have been deposited in the SRA under accession numbers (SRR14338904, SRR14338897, SRR14493697, and SRR14493696).

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
