# Peer review of "High-Molecular-Weight Plasmids Carrying Carbapenemase Genes blaNDM-1, blaKPC-2, and blaOXA-48 Coexisting in Clinical Klebsiella pneumoniae Strains of ST39"

_microorganisms, 2023, doi:10.3390/microorganisms11020459_

Round 1

Reviewer 1 Report

Review of manuscript entitled: 

"High-Molecular-Weight Plasmids Carrying Carbapenemase Genes blaNDM-1, blaKPC-2, and blaOXA-48 Coexisting in Clinical Klebsiella pneumoniae Strains of ST39" by Ekaterina S. Kuzina , Angelina A. Kislichkina , Angelika A. Sizova , Yury P. Skryabin , Tatiana S. Novikova , Olga N. Ershova , Ivan A. Savin, Olga E. Khokhlova , Alexander G. Bogun , Nadezhda K. Fursova.

Klebsiella pneumoniae is a bacteria responsible for healthcare-associated infections and belongs to the ESKAPE group of pathogens. Due to conjugative KPC plasmids, MDR Klebsiella pneumoniae strains have been increasingly reported. The problem of co-transmission of plasmids with carbapenemase genes is serious and actual in the whole world.

In this study, the authors characterize 4 closely related ST39 XDR K. pneumoniae strains from 4 patients by phenotypic methods and genotyping method - whole genome sequencing (WGS) by Illumina Miseq and MinION platforms. The patients were hospitalised in the same ward, isolates were recovered in close related time (October-November), and one of the patients was associated with intestinal carriage without symptoms of infection. The authors characterised the nucleotide sequence of chromosomes and the large and low-molecular-weight plasmids.  Nucleotide sequences were submitted to the Gene bank Database. The novelty of this research is to detect of three large plasmids of IncHI1B, IncC and IncFIB groups, these plasmids carried three carbapenemase genes: blaNDM-1, bla KPC-2 and blaOXA-48  simultaneously. One of the plasmids was a hybrid carrying two regions of the resistance genes and besides them a region of the virulence genes.

The manuscript is well written and contains all the necessary elements; phenotypic and genotypic characteristics of KP bacterial strains, the plasmid comparison figures are clear, and the conclusions and discussion are correct. The manuscript is comprehensive in terms of the topic of sequencing, perhaps too small a pool of strains tested in the existing epidemic situation.

I have not found plagiarism, the references are the latest from the last few years.
Minor correction:
Lines 96, 99: gene blaSHV-12; blaCTX-M - should be italic,
It is worth focusing on the description of the ward's epidemic situation

Author Response

Dear Reviewer,

We are very grateful to you for the positive evaluation of our manuscript.

In response to your comments, we would like to clarify the following:

Q1. The manuscript is comprehensive in terms of the topic of sequencing, perhaps too small a pool of strains tested in the existing epidemic situation.

Answer. This study is extension of previously published report [Kuzina, E.S.; Novikova, T.S.; Astashkin, E.I.; Fedyukina, G.N.; Kislichkina, A.A.; Kurdyumova, N.V.; Savin, I.A.; Ershova, O.N.; Fursova, N.K. Rectal and tracheal carriage of carbapenemase genes and class 1 and 2 integrons in patients in neurosurgery Intensive Care Unit. Antibiotics 2022, 11, 886.]. Seven-point prevalence surveys, which included 60 patients in the neuro-ICU, were conducted weekly in the period from Oct. to Nov. 2019. A total of 293 clinical samples were analyzed, including 146 rectal and 147 tracheal swabs; 344 Gram-negative bacteria isolates were collected. Among them, four extensively drug-resistant Klebsiella pneumoniae strains (sequence type ST39 and capsular type K23), simultaneously carried beta-lactamase genes, blaSHV-40 and blaTEM-1B, three carbapenemase genes, blaNDM-1, blaKPC-2, and blaOXA–48, the cephalosporinase gene blaCTX-M, and two class 1 integrons, were identified.

According to your suggestion, we modified last paragraph of Introduction section (lines 147-155).

Q2. Lines 96, 99: gene blaSHV-12; blaCTX-M - should be italic

Answer. It was corrected.

Q3. It is worth focusing on the description of the ward's epidemic situation.

Answer. The detail information about epidemic situation in the neuro-ICU was presented in previously report [Kuzina, E.S.; Novikova, T.S.; Astashkin, E.I.; Fedyukina, G.N.; Kislichkina, A.A.; Kurdyumova, N.V.; Savin, I.A.; Ershova, O.N.; Fursova, N.K. Rectal and tracheal carriage of carbapenemase genes and class 1 and 2 integrons in patients in neurosurgery Intensive Care Unit. Antibiotics 2022, 11, 886.].

Reviewer 2 Report

This study entitled” High-Molecular-Weight Plasmids Carrying Carbapenemase Genes blaNDM-1, blaKPC-2, and blaOXA-48 Coexisting in Clinical Klebsiella pneumoniae Strains of ST39” has discussed an important topic concerned with resistance to carbapenems, the last resort antibiotic for treatment MDR Enterobacteriaceae. The manuscript has been well structured and written in good language, however there are some points that authors should address.

Abstract: author should write a conclusion

line 99” blaCTX–M” should be italicized and “CTX–M” lower case

line 120: -80°C

lines 128-129  should be : “The category of multidrug resistance was determined according to the criteria of Magiorakos et al. [9].” Second, did the authors test at least two antibiotics in antimicrobial groups for Enterobacteriaceae according to Magiorakos et al., 2012?

line 130: why authors didn’t confirm identification by species- specific primers for K.pneumoniae prior of WGS analysis? Also, assessment of DNA purity and concentration should be performed before WGS analysis.

Table 1 is not cited in the text.

In table 1: replace “Strain ID” with “Information”

line 198: delete “2012”

LINE 318 “SNPs” Please write in full at the first mention.

Some typing errors should be considered:

-LINE 394: through the plasmids in Russia has been published [24, 40]. “replace ,” with “.”

-LINE 419:  of different genetic clones [23, 24].

- line 451: rearrangement

Lines 422-424 “The hospital environment in addition to human-to-human transfer, is another key of antibiotic-resistant pathogens transmission and reservoir for emergence of novel resistance combinations.” Author should add this point here “This could be confirmed by detecting plasmid-mediated colistin mcr-10 and fosfomycin fosA5 resistance genes on an IncFIB plasmid in a K. pneumoniae isolate from bovine milk.” doi: 10.3389/fmicb.2021.770813

Lines 201-202 “and one strain to colistin (Table 2).” Which mcr gene was identified in this isolate?

Author Response

Dear Reviewer,

We are very grateful to you for the positive evaluation of our manuscript.

In response to your comments, we would like to clarify the following:

Q1. Abstract: author should write a conclusion.

Answer. We added the Conclusion: The spread of K. pneumoniae strains carrying multiple plasmids conferring the resistance even to last-resort antibiotics is of great clinical concern. Additionally, we had to delete some words from the Results subsection to rich 200 words in Abstract (lines 25-33).

Q2. line 99” blaCTX–M” should be italicized and “CTX–M” lower case

Answer. It was done.

Q3. line 120: -80°C

Answer. It was corrected.

Q4. lines 128-129 should be: “The category of multidrug resistance was determined according to the criteria of Magiorakos et al. [9].” Second, did the authors test at least two antibiotics in antimicrobial groups for Enterobacteriaceae according to Magiorakos et al., 2012?

Answer. The first point was corrected. The second point: Indeed, we used antimicrobials accordingly to Magiorakos et al. recommendations.

Q5. line 130: why authors didn’t confirm identification by species- specific primers for K.pneumoniae prior of WGS analysis? Also, assessment of DNA purity and concentration should be performed before WGS analysis.

Answer. Bacterial identification was done using MALDI-TOF Biotyper instrument (Bruker, Karlsruhe, Germany). WGS analysis was used for confirmation of species identification.

Q6. Table 1 is not cited in the text. In table 1: replace “Strain ID” with “Information”

Answer. Table 1 was cited in the text just before the Table (lines 196-197). “Strain ID” was replaced with “Information”.

Q7. line 198: delete “2012”

Answer. It was done.

Q8. LINE 318 “SNPs” Please write in full at the first mention.

Answer. It was done.

Q9. LINE 394: through the plasmids in Russia has been published [24, 40]. “replace ,” with “.

Answer. It was done.

Q10. LINE 419: of different genetic clones [23, 24]

Answer. According to your suggestion, we deleted this fragment.

Q11. line 451: rearrangement

Answer. It was corrected.

Q12. Lines 422-424 “The hospital environment in addition to human-to-human transfer, is another key of antibiotic-resistant pathogens transmission and reservoir for emergence of novel resistance combinations.” Author should add this point here “This could be confirmed by detecting plasmid-mediated colistin mcr-10 and fosfomycin fosA5 resistance genes on an IncFIB plasmid in a K. pneumoniae isolate from bovine milk.” doi: 10.3389/fmicb.2021.770813

Answer. According your suggestion, we added the phrase: “This statement could be confirmed by detection of plasmid-mediated transfer of resistance genes on an IncFIB plasmid between K. pneumoniae isolates” and the reference 42.

Q13. Lines 201-202 “and one strain to colistin (Table 2).” Which mcr gene was identified in this isolate?

Answer. Though one K. pneumoniae strain was resistant to colistin, we didn’t detect mcr gene in the genome of this strain. In this case we can suppose, that two peculiar properties in the genome of this strain could be the reason of the resistance to colistin:

  • in chromosome of strain SCPM-O-B-8922 four genes, homologues to gmlABC (encode the structural modification of D-galactan I) and kfoC (the function of is unknown) in rfb cluster, have been deleted and replaced by the insertion of IS5-like element ISKpn26 family transposase; in chromosomes of other strains the rfb cluster is complete and homologous to pneumoniae lipopolysaccharide O-antigen biosynthesis gene cluster, type: O1/O2, variant 2 (LT174602);
  • in comparing to other strains, in chromosome of SCPM-O-B-8922 the gene of O-antigen ligase family protein (KIF64_00770) has been disrupted by insertion of IS5-like element ISKpn26 family transposase (KIF64_00765).

It was reported, that the polyanionic lipopolysaccharide (LPS) structure of Gram-negative bacteria, consisting of a lipid A moiety, a conserved oligosaccharide core (2-keto-3-deoxyoctonoic acid, Kdo) and an O-antigen group, is the main target of colistin (https://doi.org/10.1093/femsre/fuab049).
